# Recording Natural Head Position Using Cone Beam Computerized Tomography

**DOI:** 10.3390/s21248189

**Published:** 2021-12-08

**Authors:** Tai-Chiu Hsung, Wai-Kan Yeung, Wing-Shan Choi, Wai-Kuen Luk, Yi-Yung Cheng, Yu-Hang Lam

**Affiliations:** 1Department of Computer Science, Chu Hai College of Higher Education, Tuen Mun, Hong Kong, China; 2Faculty of Dentistry, The University of Hong Kong, Pokfulam, Hong Kong, China; ndyeung@hku.hk (W.-K.Y.); drwchoi@hku.hk (W.-S.C.); wkluka@hku.hk (W.-K.L.); yycheng@hku.hk (Y.-Y.C.); retlaw@hku.hk (Y.-H.L.)

**Keywords:** natural head position, cone beam computer tomography, medical image registration

## Abstract

The purpose of this study was to develop a technique to record the natural head position (NHP) of a subject using the scout images of cone beam computerized tomography (CBCT) scans. The first step was to align a hanging mirror with the vertical (XY) plane of the CBCT field-of-view (FOV) volume. Then, two scout CBCT images, at frontal and at sagittal planes, were taken when the subject exhibited a NHP. A normal CBCT scan on the subject was then taken separately. These scout images were used to correct the orientation of the normal CBCT scan. A phantom head was used for validation and performance analysis of the proposed method. It was found that the orientation detection error was within 0.88°. This enables easy and economic NHP recording for CBCT without additional hardware.

## 1. Introduction

Correct orientation of head provides meaningful perceptions of the maxillomandibular complex for diseases diagnoses, as well as facilitating the planning and execution of surgical and non-surgical treatments [1,2,3,4].

Anatomical landmarks and planes, such as the Frankfort horizontal and Sella–Nasion planes, have been used in determining the orientation of head. However, this method was found to have low to moderate inter-rater reliability [5]. The median difference between the estimated head position and recorded natural head position was small for roll and yaw, whereas the difference in pitch was large, exhibiting a tendency for the chin to tip more posteriorly (6.3 ± 5.2 mm). This results in less severe skeletal deformities in the anterior–posterior direction. Moreover, many patients exhibit facial asymmetry, which makes the identification of anatomical landmarks very difficult, or even impossible. Studies have found that the prevalence of dentofacial asymmetry ranges from 12% to 37% in different countries and regions (e.g., the United States, Belgium, and Hong Kong) [6,7,8,9].

Natural head position (NHP) is a reproducible head position in an upright posture [10,11], with the subject focusing on a distant point at eye level. In clinical settings, patients reproduce their own NHP by looking straight into a hanging mirror at their own eyes in a balanced position. The mirror serves as a reference for defining the patient’s head orientation. Clinical methods [12,13,14,15,16,17,18,19,20,21,22,23,24,25] have been developed to record this head position in various imaging modalities. In traditional 2D photographs/radiographs, pictures are taken of the subject, along with a hanging plumb bob as reference for reproducing patients’ NHP. In stereophotogrammetry (SP), a physical reference board has been proposed for calibrating the SP system and to capture the 3D facial mesh surface with reference to the true horizontal plane [18,19,20,21]. The accuracies achieved by traditional and stereophotogrammetry (SPNHP) methods are around ±1 and greater than ±0.1 degrees, respectively.

In cone beam computerized tomography (CBCT), the image acquisition times are long; the patient’s head may move, resulting in motion artifacts during scanning. Therefore, NHP posture has sometimes been recorded using a patented inertial motion unit (IMU) [15,16,17,18,19,20,21,22,23,24,25]. Orientation readings recorded by the IMU can be used to orientate patients’ heads when they are in an NHP. IMU systems deliver very high accuracy in recording NHP; however, additional equipment and operations are required. This limits its application in daily practice. The aim of this study was to investigate direct NHP recording in CBCT without extra hardware and procedures. We firstly investigated the orientation reproducibility in the field-of-view (FOV) of a CBCT scanner, as presented in the next section. Based on this result, we used scout images of a calibrated CBCT to record patient’s NHP. Scout images are necessary before each full CBCT scan to verify the patient’s position; thus, this operation does not introduce extra radiation exposure to patients. Each scout image requires less than 0.1 s to capture; therefore, patients can present their own NHP freely and negative effects from patient’s head movements are not significant. We then matched the CBCT volume to the orientation defined by the scout images. This is known as 3D/2D registration problem.

## 2. Materials and Methods

### 2.1. Orientation Reproducibility of CBCT Scanner

To determine the overall orientation reproducibility of CBCT scanner, a high-quality acrylic card with alignment patterns were adopted for relating physical references to CBCT volume. The list of materials and tools used in this study is as follows:Cone Beam CT scanners: Newtom GiANO, Planmeca ProMax 3D Mid;A 60 × 60 × 12 mm high-quality acrylic board with alignment patterns;A 360-degree 3-plane leveling and alignment laser (Bosch GLL3-80P, Robert Bosch GmbH, Stuttgart, Germany);NHP alignment mirror placed in front of the CBCT scanner;Two tripods, one equipped with a 3D geared head (Manfrotto 410 junior geared head), two-way focusing adjuster (Velbon Super Mag Slider) and vice clamp for mounting the orientation card (item 3). Another tripod was equipped with a simple ball head for mounting the level laser (item 3);Software: MeshLab [26], 3D slicer [27], MATLAB 2013b;A Vintage 3M Calibration Phantom skull.

The orientation card is shown in Figure 1; several dotted line markers were painted on the front (AB) and left (and/or right) edge (BC) surfaces for aligning the card horizontally with the level laser beams. The visible marker lines on the top and bottom surfaces were included to align the card with the CBCT positioning lasers in relation to the direction of the mirror. Two straight parallel unequal-length grooves (2 mm depth, 40 mm and 42 mm long) were filled with radiopaque BaSO_4_ on the top flat surface of the card, parallel to the marker line (DE), so the orientation could be captured by the CBCT scanner.

The tested CBCT scanners under were Newtom GiANO, Planmeca ProMax 3D Mid. In front of the CBCT scanner, a hanging mirror was installed for the patient to look at through their own eyes to produce a personal NHP, as shown in Figure 2, Figure 3 and Figure 4. The level laser was mounted on a tripod and placed between the hanging mirror and the CBCT scanner. The orientation reference card was firstly placed in the subject scanning position of CBCT. The position could be found by using the CBCT built-in positional laser lines. The card was then fine adjusted to align with the horizontal laser beams projecting from the level laser, as shown in Figure 1b. After the orientation card was adjusted to be horizontal in the scanning position and the level laser FG was aligned with the marker line (DE), the card and level laser were fixed, thus serving as a reference for aligning the mirror.

The mirror was then adjusted so that the direct and reflecting laser beams from the mirror overlapped, which implied that laser beam FG, which passed through points F and G, as shown in Figure 2 and Figure 3, was perpendicular to the mirror. If the CBCT position lasers accurately indicated the mid-sagittal plane of the CBCT volume, the mirror was therefore parallel to the coronal plane of the CBCT volume. The radiopaque BaSO_4_ captured in CBCT volume showed the degree of angular deviation of the mirror to the coronal plane. Before applying the CBCT scanner to capture orientation, it was necessary to determine the reproducibility of its orientation. With the aligned orientation reference card, the overall orientation reproducibility of the CBCT scanner could be found using Algorithm 1:
**Algorithm 1**Perform CBCT scans on the aligned orientation reference card N times, with the scan head starting position randomly chosen in different vertical and horizontal positions;Load each CBCT volume into 3D Slicer, and segment the orientation bars by intensity thresholding (Figure 5);Align all segmented orientation bars for all N scans in MeshLab using iterative closest point (ICP) [26] alignment (Figure 6);Calculate the Euler’s angles from the resulting alignment matrices;Calculate the means and standard deviations (SD) of the Euler’s angle deviations.

### 2.2. Orientation Correction of CBCT Volume from Scout Images in NHP

In Section 3, it is presented that the CBCT scanner can be used to record orientation, with a repeatability of within 0.17°. However, the subject is required to be stationary during scanning. The acquisition time for a normal full-skull CBCT scan is 18 s (Planmeca ProMax 3D Mid). It is difficult for a subject to maintain NHP perfectly for this time. Therefore, in this paper, it is proposed to use scout images to record the orientation when the subject is in NHP. Then, the orientation of the CBCT volume can be corrected with normal scanning settings by matching the CBCT volume to the scout images.

Let $f_{o}\left(x,y,z\right)$ be the normal CBCT scan volume and $s_{o,LR}\left(u,v\right)$ and $s_{o,AP}\left(u,v\right)$ be its lateral and frontal scout images, respectively. Furthermore, let $s_{N,LR}\left(u,v\right)$ and $s_{N,AP}\left(u,v\right)$ be the lateral and frontal scout images, respectively, when the subject is in NHP. The scout images and CBCT volume can be modeled using the Radon transform:(1)$$s_{LR}\left[f\left(x\right)\right]=\int_{-\infty }^{\infty }f\left(x,v,u\right)dx$$
(2)$$s_{AP}\left[f\left(x\right)\right]=\int_{-\infty }^{\infty }f\left(u,v,z\right)dz$$
where $x:={\left[x,y,z\right]}^{T}$.

The objective is to identify a proper rigid transformation (three-dimensional rotation and translation only) with parameters μ which move the normal volume to NHP volume [28,29,30,31],
(3)$$f_{N}\left(x\right)=f_{o}\left(T_{\mu }\left(x\right)\right)$$
where $T_{\mu }\left(x\right)=Ax+t$, A is a 3×3 rotation matrix, and t is a 3×1 translation vector. The problem is the same as the common CT volume registration, which is minimization of the objective function error:$$E\left(\mu \right)=C\left(T_{\mu };f_{o},f_{N}\right)$$

However, it is not possible to obtain $f_{N}\left(x\right)$, which is the CBCT scan when the subject is in NHP. Instead, the objective function is changed to minimize the differences between scout images. Specifically, 2D images are registered to 3D volume projections [29,30,31,32,33,34]:(4)$$E_{0}\left(\mu \right)=C\left(T_{\mu };f_{o},s_{LR}\left[f_{N}\right]\right)$$
(5)$$E_{1}\left(\mu \right)=C\left(T_{\mu };f_{o},s_{AP}\left[f_{N}\right]\right)\;$$
where C is a cost function. *T* radiation dose is typically smaller in scout image scanning; therefore, digitally rendered radiographs (DRRs) have different intensity values from the scout images. From previous studies [35,36], it was found that the angle difference between the FH plane and the true horizontal plane has a standard deviation of between approximately 4.5° and 5.6°. The reproducibility of NHP in lateral head X-rays is close to 2° [37]. Therefore, when matching the direction of the normal volume with NHP scout images, the search range of the angle deviations was set to 10°. We conjecture that the direct registration of scout images and DRR can achieve high accuracy in orientation detection. We suggest the following Algorithm 2 to find the pitch, roll and yaw angles:
**Algorithm 2**For a yaw angle $\phi_{yaw}$ within the search range:
Rotate the CBCT volume in yaw with angle $\phi_{yaw}$;Generate frontal and lateral DRR, $s_{AP}\left[f_{o}\right]$ and $s_{LR}\left[f_{o}\right]$ using Equations (1) and (2), from CBCT volume $f_{o}$;Perform 2D image registration on frontal NHP scout image $s_{AP}\left[f_{N}\right]$ and DRR $s_{AP}\left[f_{o}\right]$ to find the roll angle deviation. Obtain the optimal roll angle $\phi_{roll}$ with minimum error $E_{1}$;Perform 2D image registration on the lateral NHP scout image $s_{LR}\left[f_{N}\right]$ and DRR $s_{LR}\left[f_{o}\right]$ to find the pitch angle deviation. Obtain the optimal pitch angle $\phi_{pitch}$ with minimum error $E_{0}$;
$E\left(\mu \right):=E_{0}\left(\mu \right)+E_{1}\left(\mu \right)$, where μ consists of $\phi_{pitch}$, $\phi_{roll}$ and $\phi_{yaw}$.

The orientation angles $\phi_{pitch}$, $\phi_{roll}$ and $\phi_{yaw}$, which minimize E, will be selected.

## 3. Results

For the CBCT scanner repeatability test, scanning was performed on two days. The number of scans, N, was 20, according to the available time slot of the experiment date. Six of them were not successful due to an incorrect field-of-view. From the 14 scans, 1 was selected as a base model for the other scans to match, as shown in Figure 6. The resultant angulation deviations in pitch, roll and yaw, r.w.t., around the CBCT volume axes are shown in Table 1. The standard deviations of the angulation deviations were found to be 0.0525°, 0.1588° and 0.1683°, respectively. Compared to the ±1° and ±0.1° in registered natural head position (RNHP) [5] and SPNHP systems, this clearly showed that the device orientation reproducibility may be better than typical RNHP methods, but could not outperform the SPNHP system.

The orientation reference card can also be used as reference to compensate possible CBCT volume orientation deviations to physical verticals and the mirror. In the experiment, the means of the deviations were found to be 0.043969°, 0.1619261° and −0.041678°, respectively. These were very small because the CBCT scanner was designed to rotate in the horizontal plane.

To test the performance of the proposed Algorithm 2, a Vintage 3M Calibration Phantom skull (Figure 4a) was adopted for the simulation. This was placed in the normal CBCT scanning position. Scout images and full CBCT scans were taken. Then, the CBCT volume was rotated to a total of 125 different sets of orientation $\left\{\theta_{k}:\;\theta_{k}=\left(\theta_{pitch,k},\theta_{roll,k},\theta_{yaw,k}\right)\right\}$, from −6° to 6° in pitch, roll and yaw angles. The proposed Algorithm 2 was developed on the MATLAB platform (Mathworks Matlab 2021a) for performing CBCT volume rotation, DRR generation, image registration and result rendering (Figure 7, Figure 8 and Figure 9). For each simulated orientation $\theta_{k}$, a pair of DRRs were generated. Figure 7 shows the frontal and lateral simulated projections from the CBCT scan of a phantom head. The projections in green are the CBCT scan in original orientation, whereas those in magenta are rotated 4° and 8° pitch and roll angles, respectively. Figure 8 shows the registered projections (multimodal) using mutual information metrics with rigid transformation (Mathworks Matlab). The initial position of the moving image was calculated from the centroids of scout images. All the optimizations for the rotation combinations converged. The errors in the orientation detection results are shown in Figure 9. It was found that the detection errors were within [−0.8232°, 0.7957°], [−0.411°, 0.5642°] and [−0.35°, 0.85°] for pitch, roll and yaw angles, respectively.

For SPNHP [18,19,20,21], camera calibration and additional reference recording is needed. The reference recording was a 3D photograph taken against a marked reference board which was physically aligned vertical and parallel to the alignment mirror. Physical references were extracted automatically from the markings on the board and saved for correcting the orientation of the subsequent 3D photographs. This procedure had to be performed biweekly before taking the 3D photographs. Three-dimensional photographs of the subject could then be taken when the subject was exhibiting NHP. The subject’s NHP could be ascertained from the recorded reference and the 3D photograph. The only burden was the manual placement of reference board. For the proposed method using the CBCT scanner (CBCTNHP) in recording NHP, the installation was the NHP alignment mirror with the reference card. After installation, the CBCT scanner could be used to record NHP: two scout images were taken when the subject was exhibiting NHP; then, a normal CBCT scan was conducted. The subject’s NHP could then be found from registering the CBCT volume with the scout images. Modern CBCT scanners offer automatic calibration; therefore, NHP recording with CBCT scanners is much easier to perform. The 2D–3D image registration is only necessary when many efficient algorithms are readily available [29,30,31,32,33,34,35,36].

## 4. Conclusions

In this study, we developed a technique to record references and correct the orientation of CBCT volume for analyzing natural head position (NHP) using CBCT scout images. A reference card was used to record physical references for quantifying the orientation reproducibility of CBCT scanners. It showed that current CBCT scanners have very high orientation reproducibility (within ±0.17°); hence, can be applied for recording NHP.

To ensure the subject maintains their own NHP, it is proposed to make use of two scout CBCT images when the subject exhibits NHP as references for correcting the CBCT volume taken in normal settings. A phantom head was used for validation and performance analysis of the proposed method. The performance was comparable to existing RNHP methods (except SPNHP, within ±0.1°) when limiting the rotation to within ±6°, and the correction error down to within 0.88° and 0.37° in pitch and roll, respectively. Currently, registration of 2D scout images with the CBCT volume is implemented based on intensity images. Generating DRRs is needed in the orientation searching range; thus, calculation of the cost function is expensive, as is the execution. More sophisticated registration methods are under investigation to improve the orientation detection performance, such as including the gradient DRR images and feature point detection in addition to the intensity images.

Recording natural head position with the scout views of CBCT enables many applications, such as simulations of jaw movement and enabling computerized planning in orthognathic and prosthetic surgery. This method is simple, does not require expensive equipment, and involves no additional radiation exposure to the patients.

## Figures and Tables

**Figure 1 sensors-21-08189-f001:**
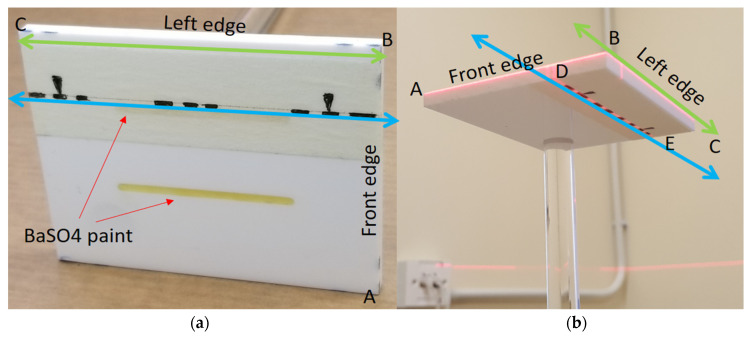
(**a**) Orientation reference card: a high-quality acrylic card with alignment patterns on its edges (green lines) and top and bottom surfaces (blue lines). On the top surface, two grooves were made and filled with radiopaque BaSO_4_. (**b**) Scanning position with the edges aligned to the horizontal plane with a level laser. The mirror was further aligned with the marker line DE on the top or bottom surfaces with direct and reflected laser beams.

**Figure 2 sensors-21-08189-f002:**
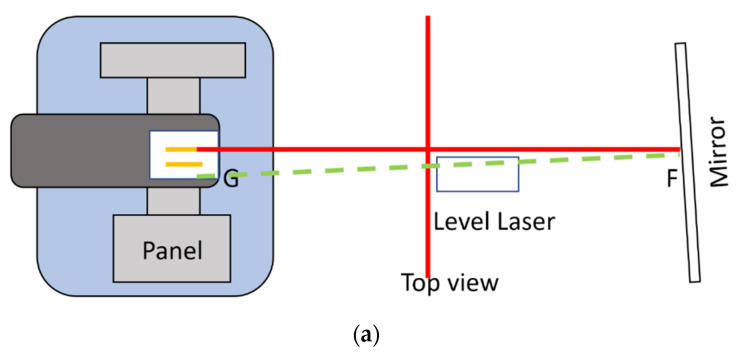
(Top view) The level laser was firstly aligned with the marker lines on the top or bottom surfaces of the orientation card. Red lines indicate the top view of direct laser beams of the two vertical laser planes. The green line indicates the laser beams reflected the from mirror. (**a**) This shows that the mirror is not yet aligned with the orientation card, because the red and green lines do not overlap. (**b**) The red and green lines overlap; therefore, the mirror is now aligned with the orientation card, and perpendicular to the laser beam FG.

**Figure 3 sensors-21-08189-f003:**
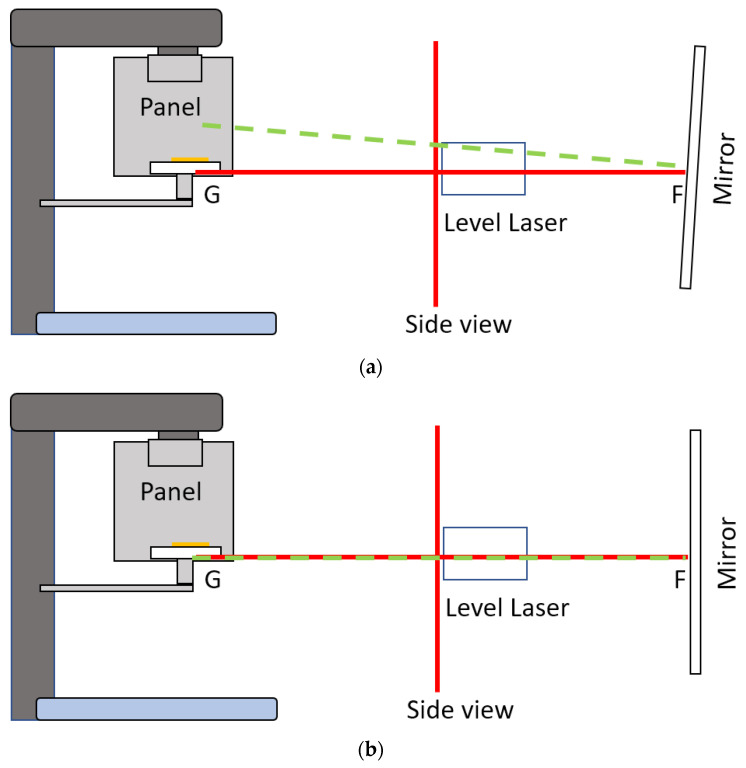
(Side view) Alignment of the mirror with the orientation card. The red line represents the side view of the projected laser beam in the horizontal and one of the vertical laser planes. The green line represents the laser beam reflected by the mirror. (**a**) The red and green lines do not overlap; therefore, the tilt angle of the mirror needs to be adjusted. (**b**) The red and green lines overlap; therefore, the mirror is now aligned with the orientation card and perpendicular to the laser beam FG.

**Figure 4 sensors-21-08189-f004:**
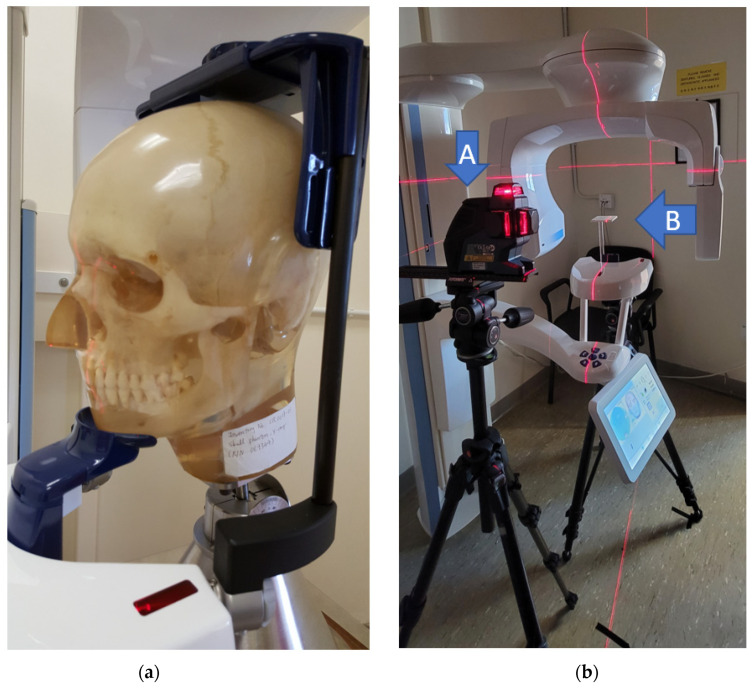
(**a**) Vintage 3M Calibration Phantom skull used in the simulation. (**b**) Planmeca CBCT scanner without the head-stabilizing parts. The level laser (A) is aligned with the orientation card (B) and was further used to align the mirror.

**Figure 5 sensors-21-08189-f005:**
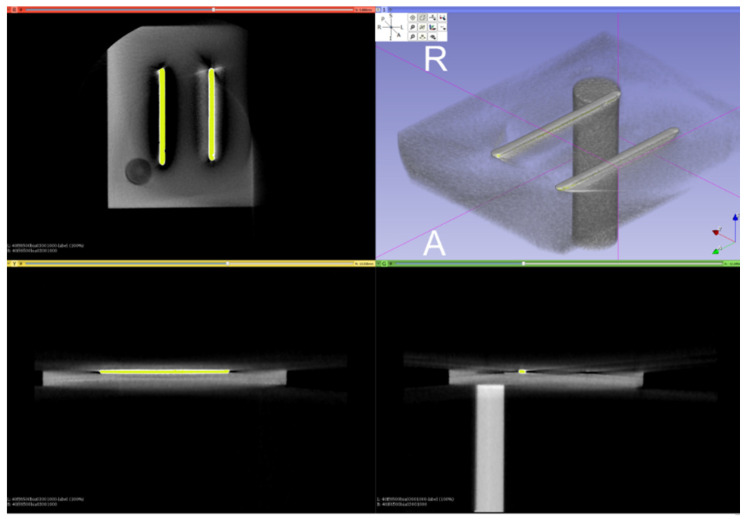
Segmentation of the orientation bars from the CBCT scan of the card using intensity thresholding.

**Figure 6 sensors-21-08189-f006:**
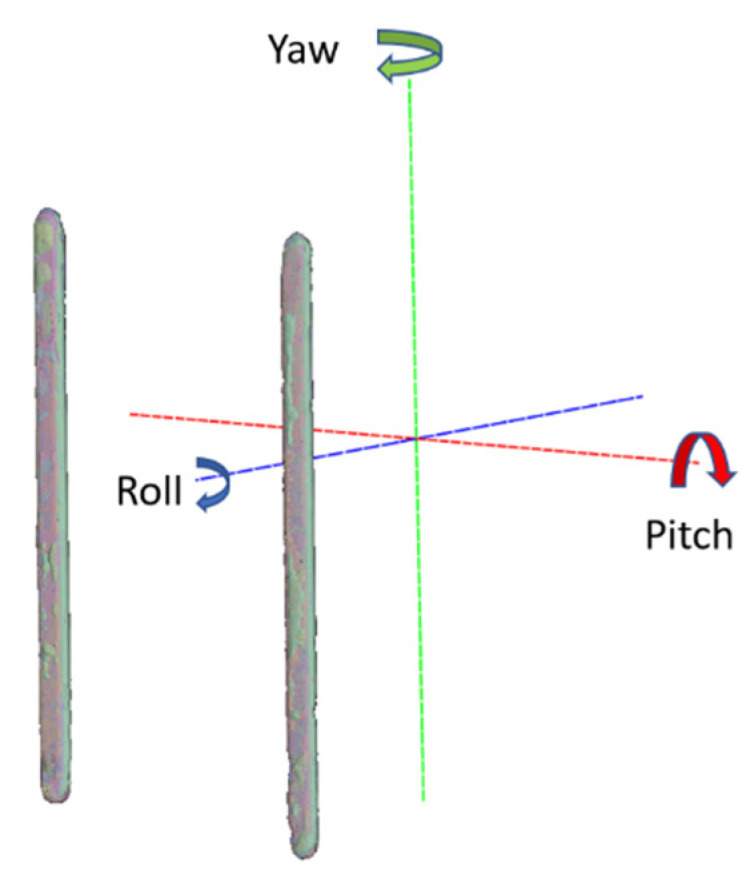
ICP alignment results of the orientation bars.

**Figure 7 sensors-21-08189-f007:**
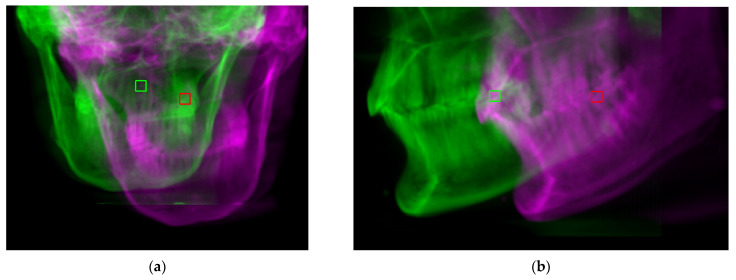
Digitally rendered radiographs from a normal CBCT scan in original (green) orientation and rotated −6 and 6 degrees in pitch and roll, respectively (magenta). (**a**) Frontal. (**b**) Lateral.

**Figure 8 sensors-21-08189-f008:**
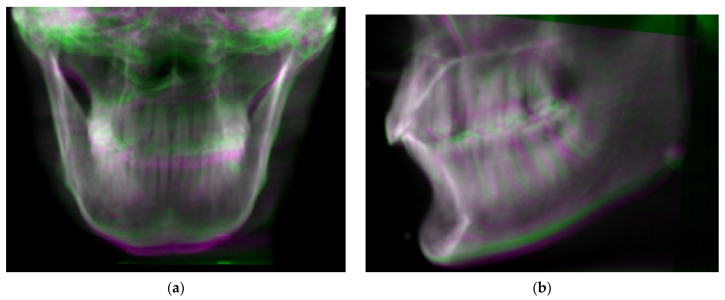
Simulated projections from normal CBCT scans in original (green) orientation and rotated −6 and 6 degrees in pitch and roll, respectively (magenta). (**a**) Frontal. (**b**) Lateral.

**Figure 9 sensors-21-08189-f009:**
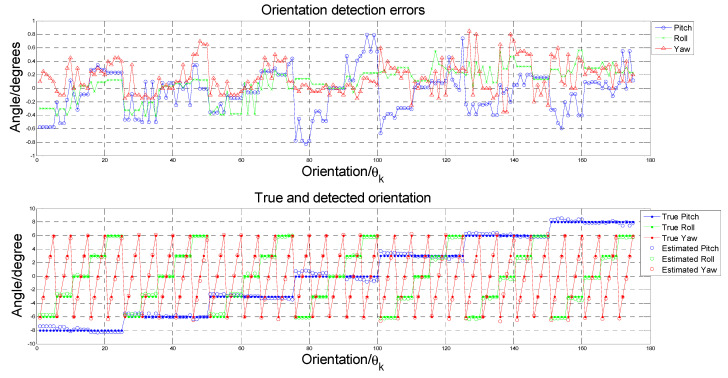
**(****Upper**) Orientation detection errors for each orientation $\theta_{k}$. (**Lower**) the true and detected angles for each orientation $\theta_{k}$.

**Table 1 sensors-21-08189-t001:** Angulation deviation in pitch, roll and yaw, in degrees, for 14 CBCT scans of the orientation card.

Pitch	Roll	Yaw
0.085694	0.031787	−0.176710
0.069415	−0.005070	−0.238110
0.077497	0.070634	−0.229910
0.073371	0.022850	−0.224360
0.079104	0.102250	−0.132350
−0.024020	0.425442	0.085904
0.045774	0.331823	0.207635
0.027183	0.457743	0.179157
−0.092900	0.341714	0.233181
0.060639	0.147581	−0.002380
0.107892	0.115700	−0.033350
0.061695	0.099723	−0.042980
0.044223	0.124786	−0.209220

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
