# Peer review of "Recording Natural Head Position Using Cone Beam Computerized Tomography"

_sensors, 2021, doi:10.3390/s21248189_

Round 1

Reviewer 1 Report

Dear Authors

At the outset, thank you for undertaking interdisciplinary research of great importance from the point of view of the development of diagnostic methods in the craniofacial area. A very interesting article, written with great scientific solidity. However, I suggest introducing a few modifications that will make the article more readable and accessible

1)      Expand point 3. Results - make a careful and comprehensive description of individual figures and specify the diagnostic modifications used (Figure1 - Figure 6), e.g. obtaining repeatability of the tests and the number of tests performed during the experiment.

2)      Please make a broader analysis of the results (Figure 7 - Figure 9),

3)      Please shorten the discussion and transfer some descriptions to appropriate drawings - easier analysis of the results obtained,

4)      In chapter 4. Discussion, please focus on the analysis of your own research results, there is no need to refer to the literature part. I propose to separate this point and transfer one part directly under the results and leave the summary part.

5)      The charts of Figure 9 should be numbered according to the form of the article (a, b) or another number of Figure may be given. The legend in the charts is not legible - please adjust the font so that it is legible without the need for a very large magnification.

6)      I propose to enrich the article with conclusions in which the results of the research, possibilities of diagnostic applications and project evaluation will be summarized.

7)      I suggest enriching the literature review with more up-to-date items. Only 2 literature items concern publications from the last 5 years.

Once again, congratulations on taking up such an interesting and ambitious topic.

Reviewer 2 Report

I congratulate for the manuscript!

My comments for the corrections are the following:

In Materials and Methods from line 64 as a "materials and tools used in this study" the head phantom is missing and its properties as well. It needs to be clarified.

In line 83 "....the laser beam FG is perpendicular to the mirror." was written.  FG as an abbreviation shall be explained.

In line 93 and 94 "...for 14 times, with 93 the scan head starting position randomly chosen in different vertical and horizontal positions." It shall be clarified how the number of scans (14) was determined and how the different positions were randomly determined?

In line 107 explanation of abbreviation "RNHP " is missing. It needs to be explained!

Location of Table 1 shall be revised, if it is containing results of the current study. In the manuscript it can be found in Materials and Methods.

In line 120 "normal full skull CBCT scan is 18 sec" is stated. Please clarify, in case of which CBCT appliance?

In line 118-119 "From Section 2, it is shown that the CBCT scanner can be used to record orientation 118 with repeatability within 0.17°." is stated. The statement shall  be clarified, namely the statement is based on a result of the present study or it is steted based on a previous publication?

In line 158-159 " In the application of matching orientation of normal volume to NHP scout images, the range of orientation deviation is typically less than 10°." This statement is not confirmed by reference or any results of the present study in the manuscript. Please, clarify the statement!

In Materials and methods please clarify, that in which software platform was the Algorithm 2 implemented to? 

The software from which the image of Figure 7 was taken shall be clarified. Only in case of Figure 8 it was stated in line 187 "(Mathworks Matlab)."

I'm ready to reconsider after major revision.

Round 2

Reviewer 2 Report

I accept the revised manuscript and I congratulate for the well written manuscript!